# BES1 is activated by EMS1-TPD1-SERK1/2-mediated signaling to control tapetum development in *Arabidopsis thaliana*

Weiyue Chen[1,2], Minghui Lv[1,2], Yanze Wang[1], Ping-An Wang[1], Yanwei Cui[1], Meizhen Li[1], Ruoshi Wang[1], Xiaoping Gou[1] & Jia Li[1]

BES1 and BZR1 were originally identified as two key transcription factors specifically regulating brassinosteroid (BR)-mediated gene expression. They belong to a family consisting of six members, BES1, BZR1, BEH1, BEH2, BEH3, and BEH4. *bes1* and *bzr1* single mutants do not exhibit any characteristic BR phenotypes, suggesting functional redundancy of these proteins. Here, by generating higher order mutants, we show that a quintuple mutant is male sterile due to defects in tapetum and microsporocyte development in anthers. Our genetic and biochemical analyses demonstrate that BES1 family members also act as downstream transcription factors in the EMS1-TPD1-SERK1/2 pathway. Ectopic expression of both *TPD1* and *EMS1* in *bri1-116*, a BR receptor null mutant, leads to the accumulation of non-phosphorylated, active BES1, similar to activation of BES1 by BRI1-BR-BAK1 signaling. These data suggest that two distinctive receptor-like kinase-mediated signaling pathways share BES1 family members as downstream transcription factors to regulate different aspects of plant development.

---

[1] Ministry of Education Key Laboratory of Cell Activities and Stress Adaptations, School of Life Sciences, Lanzhou, China. [2]These authors contributed equally: Weiyue Chen, Minghui Lv. Correspondence and requests for materials should be addressed to J.L. (email: lijia@lzu.edu.cn)

 

Brassinosteroids are essential plant hormones regulating multiple processes during plant growth, development, and stress adaptations[1–4]. Brassinosteroids (BRs) are perceived by a cell surface RLK complex, including their major receptor BRASSINOSTEROID INSENSITIVE 1 (BRI1) and co-receptor BRI1-ASSOCIATED RECEPTOR KINASE 1 (BAK1)[5–8]. Upon the activation of BRI1 and BAK1[9–11], an intracellular signaling cascade can be subsequently initiated[12], including releasing a negative regulator BRI1 KINASE INHIBITOR 1 (BKI1) from the plasma membrane-localized BRI1 and BAK1 complex[13–15], and inactivating second negative regulator BRASSINOSTEROID INSENSITIVE 2 (BIN2) by a phosphatase BRI1 SUPPRESSOR 1 (BSU1)[12,16–19]. Inactivation of BIN2 can greatly induce the accumulation of non-phosphorylated forms of two transcription factors, BRI1 EMS SUPPRESSOR 1 (BES1) and BRASSINAZOLE RESISTANT 1 (BZR1), in the nucleus[20–24]. Activated BES1 and BZR1 can then mediate the expression of thousands of downstream responsive genes by either directly associating with the promoters of their target genes or via interacting with different types of transcription factors[25–27].

It has been believed that BES1 and BZR1 are essential transcription factors in the BR signaling pathway to regulate target gene expression[20,21]. Loss-of-function genetic data supporting the importance of BES1 family members in the BR signaling pathway, however, are not fully established[28]. To investigate the genetic significance of BES1 family members in the BR signaling pathway, we isolated single mutants for all BES1 family genes and generated double, triple and higher order mutants. Interestingly, our quintuple mutants showed a complete male sterile phenotype due to lack of pollen grains in its anthers, suggesting that BES1 family members are also involved in pollen generation.

Pollen grains are produced by anthers, the upper part of the male reproductive organ called stamen. The development of anthers in Arabidopsis involves 14 artificially divided stages according to previous morphological and cellular studies[29–31]. Anther primordium, consisting of three layers L1, L2, and L3, emerges from floral meristem at stage 1. During stages 2–5, cells from these three layers divide and differentiate to produce a fundamental anther architecture with four lobes which are connected by L3-derived vascular tissues and are covered by L1-derived epidermis. At the meantime, L2-derived cells within each of the four lobes develop into microsporocytes at the center and three surrounding somatic cell layers including tapetum, middle layer, and endothecium from the inner to outer. At stages 6–12, microsporocytes initiate meiotic division to generate microspores that further develop into mature pollen grains. Then, mature pollen grains are released at stage 13 and anthers finally collapse at stage 14.

During anther morphogenesis, cell-to-cell communications are critical to the coordinated development of L2-derived layers such as microsporocytes and their nurturing somatic layer, tapetum[30]. Previous studies indicated that microsporocytes and their precursor cells (sporogenous cells) guide tapetum formation and differentiation via an RLK-mediated transmembrane signaling pathway[32–34]. Sporogenous cells and microsporocytes secret TAPETUM DETERMINANT 1 (TPD1), a small cysteine-rich peptide ligand, which can bind to and activate its receptor and co-receptor, EXTRA MICROSPOROCYTES1/EXTRA SPOROGENOUS CELLS (EMS1/EXS) and SOMATIC EMBRYOGENESIS RECEPTOR-LIKE KINASES 1 and 2 (SERK1 and 2)[32,35–40]. EMS1 is mainly localized in the inner secondary parietal layer (precursor of tapetum) and tapetum during anther patterning[32]. Although recent studies suggested that carbonic anhydrases act as direct targets of EMS1 in regulating tapetal cell differentiation, how activated TPD1-EMS1/SERK1/2 complex triggers downstream response gene expression is not fully understood[41].

Here we demonstrate that BES1 family members play key roles in anther development. Anthers of a quintuple mutant, bes1-1 bzr1-1 beh1-1 beh3-1 beh4-1 (qui-1), show an incompletely developed tapetum-like cell layer, which losses normal tapetum cell identity. In addition, qui-1 displays excessive microsporocytes, similar to what is observed in ems1, tpd1, serk1/2 mutant anthers. Gain-of-function point mutation of BES1 or BZR1, bes1-D or bzr1-1D, can partially restore the tapetum identity and pollenless defects of ems1, tpd1, and serk1/2. Consequently, the fertilities of these mutants are considerably recovered. Ectopic expression of TPD1 and EMS1 leads to significant accumulation of non-phosphorylated BES1. Our data reveal that BES1 family members are the downstream transcription factors of the TPD1-EMS1/SERK1/2 signaling pathway and they play an essential role in plant reproduction via regulating tapetum development.

## Results

**BES1 family members are required for pollen production.** BES1 and BZR1 are two well-characterized transcription factors originally identified as downstream components of the BR signal transduction[20,21]. Loss-of-function mutants of BES1 or BZR1 do not show typical BR-related defects. Phylogenic analysis indicated that BES1 and BZR1 belong to a six-member family, including BES1, BZR1, BEH1, BEH2, BEH3, and BEH4 (Supplementary Fig. 1). We therefore ordered T-DNA insertion lines for all of the BES1 family member genes from Arabidopsis Biological Resource Center (ABRC). Genotypic analyses confirmed the true T-DNA insertions for BES1, BZR1, BEH1, BEH3, and BEH4, but not for BEH2. These five T-DNA insertion lines were subsequently named as bes1-1, bzr1-1, beh1-1, beh3-1, and beh4-1, respectively (Supplementary Fig. 2a). RT-PCR analyses proved that bzr1-1, beh1-1, and beh3-1 are true null mutants, in which full-length mRNAs cannot be detected (Supplementary Fig. 2b). Previous studies indicated that BES1 has two alternatively spliced transcripts, BES1-L and BES1-S[42]. In bes1-1, the BES1-L transcript is completely eliminated, whereas the BES1-S transcript is still present (Supplementary Fig. 2b). Therefore, bes1-1 is not a complete knockout line, even though it was demonstrated that BES1-L is more functionally important than BES1-S[42]. beh4-1 is actually a knockdown mutant but not a null mutant (Supplementary Fig. 2b). All these single mutants did not show any growth defects. We then generated different combinations of double, triple, and quadruple mutants. Double, triple, and quadruple mutants generated from these T-DNA alleles did not display obvious phenotypes. A quintuple mutant, bes1-1 bzr1-1 beh1-1 beh3-1 beh4-1 (qui-1) showed a complete sterile phenotype (Fig. 1a, b and Supplementary Fig. 3). Filaments of qui-1 are much shorter than that of wild type and fail to elongate (Fig. 1c, d). Microscopic assays indicated that the sterility is caused by lack of pollen grains in the anthers of the quintuple mutant (Fig. 1e–h and Supplementary Fig. 3g–l). To further confirm that the male sterility is caused by the inactivation of the five BES1 family genes, we introduced the genomic sequences of five genes, mostly driven by their own promoters, into qui-1 individually (Supplementary Fig. 4). Cloning of BEH3 promoter was unsuccessful after multiple trials due to unknown reasons. UBQ10 promoter was therefore used to drive the expression of BEH3 (Supplementary Fig. 4). All the transgenic plants obtained showed fully recovered pollen development and fertility (Supplementary Fig. 4). These results indicated that the male sterility of the qui-1 is solely caused by the inactivation or partially inactivation of five BES1 family genes. qui-1 is not a complete knockout line and it does not show any obvious vegetative growth defects besides its male sterility. Therefore, most of our subsequent analyses were performed by using qui-1.

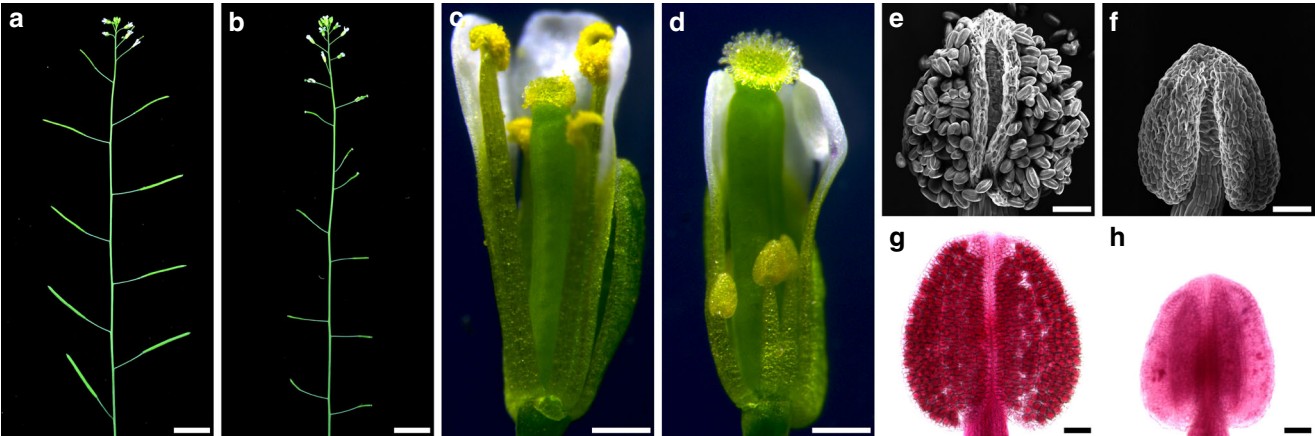

**Fig. 1** A quintuple mutant, *bes1-1 bzr1-1 beh1-1 beh3-1 beh4-1* (*qui-1*), shows a male sterile phenotype due to shortened filaments and pollenless anthers. **a**, **b** Inflorescences phenotypes of wild-type Col-0 (**a**) and *qui-1* (**b**). Col-0 siliques are elongated and produce viable seeds, while mutant siliques fail to elongate and are devoid of any seeds. **c**, **d** Optical micrographs of Col-0 (**c**) and *qui-1* (**d**) flowers. *qui-1* develops stamens with filaments much shorter than Col-0. **e**–**h** Anthers of Col-0 (**e**, **g**) and *qui-1* (**f**, **h**) visualized by a SEM microscope (**e**, **f**) and Alexander staining (**g**, **h**), respectively. Col-0 anthers give rise to viable pollen grains while mutant anthers are completely pollenless. Scale bars represent 1 cm in (**a**), (**b**), 0.5 mm in (**c**), (**d**), and 50 μm in (**e**–**h**)

**BES1 family members are essential for tapetum development.**
To investigate the cellular basis causing pollen developmental defects, we made a series of semithin anther cross sections. We then compared the anatomical structures of anthers from Col-0 and *qui-1* from stages 4–7 (Fig. 2). At stage 4 and early stage 5, the anther structures from Col-0 and *qui-1* look indistinguishable (Fig. 2a, b, g, h). The structural differences can be clearly observed from late stage 5, when the tapetum cells in Col-0 start to enlarge and differentiate (Fig. 2c), cells at the same position remain unaltered in *qui-1* (Fig. 2i). Meanwhile, numbers of microsporocytes in *qui-1* start to increase and partially crush the undeveloped tapetum-like cell layer (Fig. 2i). At stage 6, Col-0 tapetum cells further enlarge and differentiate (Fig. 2d, e), whereas tapetum-like cells in *qui-1* fail to enlarge and differentiate (Fig. 2j, k). At the meantime, the volume and cell numbers of microsporocytes are quickly increased in *qui-1* but not in Col-0 (Fig. 2d, e, j, k). At stage 7, Col-0 tapetum cells start to degenerate and tetrads are formed (Fig. 2f). In *qui-1*, microsporocytes cannot further develop into tetrads and cell death is initiated (Fig. 2l). These observations indicated that the major reason causing the pollenless anthers in *qui-1* is likely the abnormally developed tapetum.

To examine whether the tapetum-like cell layer in *qui-1* losses tapetum cell identity, we carried out an in situ hybridization analysis to check the expression levels of two known tapetum marker genes, *DYSFUNCTIONAL TAPETUM1* (*DYT1*)[43] and *ARABIDOPSIS THALIANA ANTHER 7* (*ATA7*)[44] (Fig. 3a–f). At stage 6 and stage 8, the expression of *DYT1* and *ATA7* can be, respectively, observed in the tapetum cell layer of Col-0 (Fig. 3a, d). In *qui-1*, however, the expression levels of these two genes are largely downregulated (Fig. 3b, e). This result is consistent with the quantitative PCR analysis (Fig. 3g). In addition, we also compared the expression levels of a number of tapetum identity related genes, including *A6*[45], *A9*[46], *TDF1*[47], *AMS*[48,49], *MS1*[50], *MS2*[51], and *MYB103*[52], in Col-0 and *qui-1*. In *qui-1*, the expression levels of these selected genes are largely suppressed (Fig. 3g). These data support our hypothesis that the tapetum cell identity in *qui-1* has been altered.

**BES1 and BZR1 are expressed in anthers.** If BES1, BZR1, and other family members are involved in the development of tapetum cells, they should be detected in the tapetum. To test this hypothesis, we successfully introduced *pBES1::gBES1-YFP*,

*pBZR1::gBZR1-YFP*, *pBEH1::gBEH1-YFP*, and *pBEH4::gBEH4-YFP* into Col-0 and analyzed the localization of these proteins in anthers. Although they are not tapetum-specific proteins, tapetum localization of these four proteins can be clearly observed (Fig. 4 and Supplementary Fig. 5a–f). Due to technical difficulties, we failed to clone *pBEH2::gBEH2-YFP* and *pBEH3::gBEH3-YFP*. To examine whether *BEH2* and *BEH3* are also expressed in the flowers, we generated transgenic plants carrying *pBEH2::GUS* or *pBEH3::GUS* and analyzed their expression patterns. Our GUS staining results indicated that both *BEH2* and *BEH3* are indeed expressed in the floral buds (Supplementary Fig. 5g–j). The protein localization or gene expression results are consistent with our morphological analyses of *qui-1*.

**The BES1 family acts independently of BR in tapetum development.** BES1 and BZR1 were thought to be the central regulators in the BR signal transduction, by mediating the expression of thousands of downstream BR response genes[20,21,26,27]. To investigate whether blocking the BR signaling pathway can cause similar tapetum defects as seen in *qui-1*, we compared anthers from a *BIN2* gain-of-function mutant, *bin2-1*[16], two null *BRI1* mutants, *bri1-701*[53] and *bri1-116*[5], a triple BR receptor mutant, *bri1-701 brl1 brl3*, and a null mutant of a BR biosynthetic gene *CPD*, *cpd*[54], with those from Col-0 and *qui-1* by scanning electronic microscopy (Supplementary Fig. 6a). Consistent with the results from a previous report[55], all the BR-related mutants can produce pollen grains in their anthers, although they are only partially functional (Supplementary Fig. 6a). *qui-1*, on the other hand, cannot produce any pollen grains (Fig. 1e–h and Supplementary Fig. 6a). Furthermore, we made semithin sections of anthers at different developmental stages for all aforementioned mutants and Col-0. As previously reported[55], all the BR-related mutants contain a functional tapetum cell layer, although these cells are slightly more vacuolated (Supplementary Fig. 6b). These data suggest BES1 family members have additional function besides their roles in regulating BR signaling.

***qui-2* anther phenotypes suggest link to TPD1-EMS1 signaling.** To investigate whether known tapetum regulatory components can regulate the activity of BES1 family members, we compared phenotypes of several published mutants with our quintuple mutants. We found that the defective anther phenotype of *qui-1* is much closer to but slightly weaker than that of *ems1*, *tpd1*, and

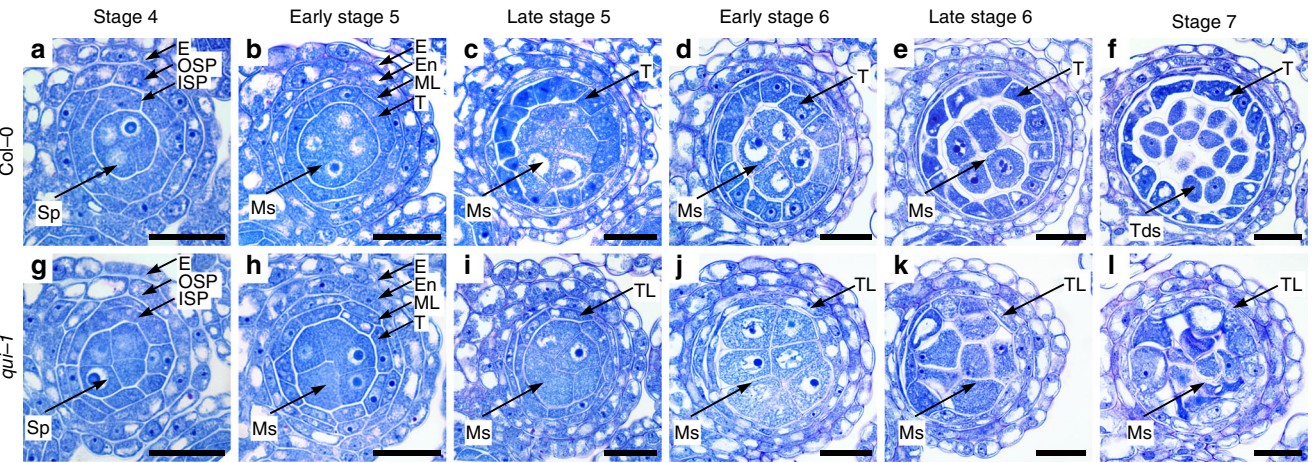

**Fig. 2** *qui-1* shows impaired tapetum differentiation and microsporogenesis starting from late stage 5. Semithin sections of anther lobes at different developmental stages from Col-0 (**a–f**) and *qui-1* (**g–l**) after stained with toluidine blue. **a**, **g** Stage 4 anthers. The anthers from Col-0 and *qui-1* showed no visible differences. **b**, **h** Early stage 5 anthers. The anthers from Col-0 and *qui-1* showed no visible differences. **c**, **i** Late stage 5 anthers. Failed enlargement of tapetal cells and excessive microsporocytes were observed in *qui-1* anthers. **d**, **j** Early stage 6 anthers. Significant excessive microsporocytes were observed in mutant anther locules. **e**, **k** Late stage 6 anthers. **f**, **l** Stage 7 anthers. Microsporocytes in *qui-1* anthers cannot form tetrads. Epidermis (E), outer secondary parietal cell layer (OSP), inner secondary parietal cell layer (ISP), sporogenous cell (Sp), middle layer (ML), tapetum layer (T), tapetum-like layer (TL), microsporocytes (Ms), and tetrads (Tds) are indicated by arrows. Scale bars represent 20 μm

*serk1 serk2* (Fig. 5a–d). The major morphological difference is that *ems1, tpd1,* and *serk1/2* completely lack a tapetum cell layer (Fig. 5d), whereas *qui-1* still has a tapetum-like cell layer (Fig. 5d), although it lacks tapetum cell identity. Because *qui-1* is leaky for *BES1* and *BEH4*, we generated additional mutants for *BES1* and *BEH4, bes1-c1, bes1-c2,* and *beh4-c1,* by using CRISPR-Cas9. Sequence analyses indicated that *bes1-c1* contains a consecutive 2-base pair (bp) mutation and two additional insertions of 8-bp and 24-bp, resulting in a premature stop codon (Supplementary Fig. 7). *bes1-c2* and *beh4-c1* contain 97-bp and 80-bp deletions, respectively (Supplementary Fig. 7). Because we did not obtain T-DNA insertion lines for *BEH2,* we also generated a CRISPR-Cas9 mutant for *BEH2, beh2-c1,* in which a 28-bp deletion was identified by sequencing analysis (Supplementary Fig. 7). Therefore, the newly generated CRISPR-Cas9 mutants are considered null mutants. Using these new mutants and previous T-DNA null mutants for *BZR1, BEH1,* and *BEH3,* we subsequently generated a second set quintuple mutant, *bes1-c1 bzr1-1 beh1-1 beh3-1 beh4-c1* (*qui-2*), a quadruple mutant, *bes1-c1 bzr1-1 beh3-1 beh4-c1* (*qua-new*), and a sextuple mutant, *bes1-c1 bzr1-1 beh1-1 beh2-c1 beh3-1 beh4-c1.* Similar to *qui-1, qui-2* does not contain any pollen grains and is completely male sterile (Fig. 5a–c and Supplementary Fig. 8). Semithin section analysis revealed that 60.2% anthers of *qui-2* are identical to those of *ems1, tpd1,* and *serk1/2,* in which the tapetum cell layer is absent (Fig. 5d and Supplementary Fig. 8h). In *qui-2,* 39.8% anthers only contain undifferentiated somatic cells, similar to those from *spl* (Supplementary Fig. 8l). Unlike *qui-1, qui-2* shows a dwarfed stature (Supplementary Fig. 8a–d), suggesting the BR signaling pathway has also been partially damaged. In *bes1-c1 bzr1-1 beh1-1 beh2-c1 beh3-1 beh4-c1* sextuple mutant, 12.7% anthers contain 2–3 somatic cell layers surrounding the microsporocytes, and 87.3% anthers show a *spl*-like phenotype (Supplementary Fig. 8e, i, m). Unsurprisingly, the tapetum defects of *bes1-c1 bzr1-1 beh3-1 beh4-c1* is more severe than those of *qui-1* but less severe than *qui-2* (Supplementary Fig. 8f–h, j–l). We also found the expression levels of tapetum marker genes, *A6, A9, ATA7,* and *DYT1,* are greatly downregulated in *qui-1,* similar to those in *ems1, tpd1, serk1/2* (Fig. 5e). These genetic and molecular results strongly suggest BES1 family members act in the same signaling pathway

with EMS1, TPD1, and SERK1/2 to control tapetum development.

**BES1 family members act downstream of TPD1-EMS1/SERK1/2.** To confirm BES1 family members are in the same signaling pathway of TPD1-EMS1-SERK1/2, we introduced gain-of-function mutation of *BES1* or *BZR1, bes1-D* or *bzr1-1D,* into *tpd1, ems1,* and *serk1/2* independently by genetic crossing (Fig. 6 and Supplementary Fig. 9). Genotypic and phenotypic analyses indicated the *bes1-D* and *bzr1-1D* can partially restore a functional tapetum cell layer, pollen development, and fertility in *tpd1, ems1,* and *serk1/2* (Fig. 6 and Supplementary Fig. 9). *bes1-D bzr1-1D* double mutation showed better restoration to *tpd1* than *bes1-D* or *bzr1-1D* single mutation (Fig. 6). Consistently, we found that the expression levels of several tapetum marker genes, *A6, A9, ATA7,* and *DYT1,* are greatly downregulated in *tpd1,* and replacing *BES1* with *bes1-D,* or *BZR1* with *bzr1-1D,* or both *BES1* and *BZR1* with *bes1-D* and *bzr1-1D,* by genetic crossing can significantly recover the expression of these tapetum marker genes (Fig. 6p). Different from their roles in *tpd1, ems1,* and *serk1/2, bes1-D* and *bzr1-1D* cannot restore pollen development and fertility of *dyt1*[43], another mutant with defects in tapetum function (Supplementary Fig. 10). We also replaced *BES1* or *BZR1* with *bes1-D* or *bzr1-1D* in a number of other anther morphological mutants, including *spl/nozzle*[56,57], *bam1/2*[58], and *rpk2*[59], no obvious fertility has been recovered (Supplementary Fig. 10). Our quantitative RT-PCR results indicated that the incapability of *bes1-D* or *bzr1-1D* to suppress the sterile phenotype of *spl/nozzle, bam1/2,* and *rpk1* is unlikely due to the low expression levels of *bes1-D* or *bzr1-1D* in these mutants (Supplementary Fig. 11). It is worth mentioning that 13.8% (*n* = 80) lobes from *spl bes1-D* and 4.4% (*n* = 68) lobes from *spl bzr1-1D* showed pollen-like structures by semithin section analyses, although both double mutants are still male sterile (Supplementary Fig. 12). These data indicated that the function of *bes1-D* and *bzr1-1D* in recovering the male reproduction defects of *tpd1, ems1,* and *serk1/2* is mutant-specific.

If BES1 and its paralogs are the downstream components in TPD1-EMS1-SERK1/2 signaling pathway, we expect the activated ligand-receptor-co-receptor complex can activate BES1 and its

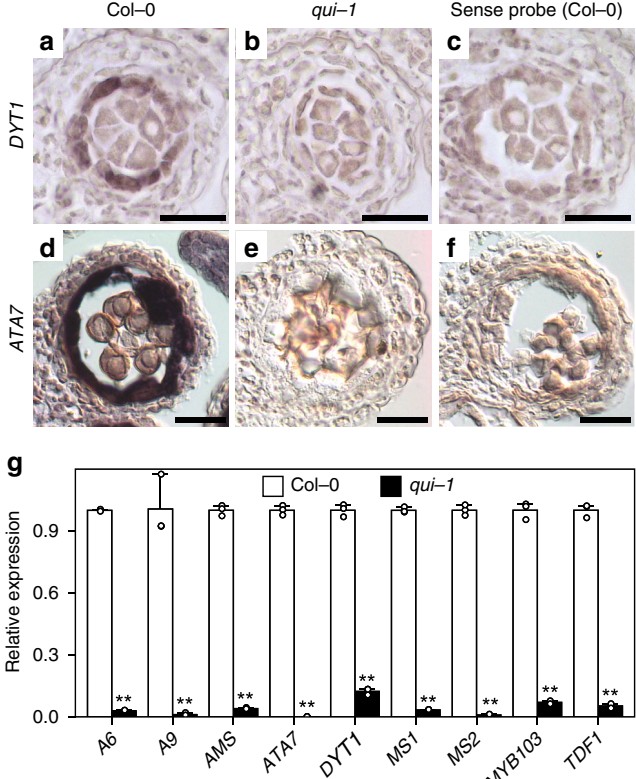

Fig. 3 BES1 family members are essential to tapetum development and function. **a–f** In situ hybridization analyses of *DYT1* (**a–c**) and *ATA7* (**d–f**) in the anthers of Col-0 (**a, d, c, f**) and *qui-1* (**b, e**). Hybridization signals can be detected in stage 6 and stage 8 anthers of Col-0 (**a, d**), but not in the same-stage anthers of *qui-1* (**b, e**), respectively. Sense probe was used as negative controls (**c, f**). Scale bars represent 20 μm. **g** Quantitative PCR analysis results to show relative expression levels of several tapetum marker genes in the inflorescences of Col-0 and *qui-1*. Data are presented as mean and s.d. (*n* = 3). Asterisks indicate significant differences (*P* < 0.01, two-tailed *t*-test)

paralogs. When BR signaling is initiated, non-phosphorylated BES1 can be greatly accumulated. We therefore tested whether similar consequence can occur. To exclude the interference of the BR signaling pathway, we also generated transgenic plants harboring *pBRI1::TPD1* and *pBRI1::EMS1-GFP* in *bri1-116*, individually. We then created double transgenic plants in Col-0 and in *bri1-116* by genetic crossing. We tested the phosphorylation levels of BES1 with or without the treatment of brassinolide (BL), the final product of the BR biosynthetic pathway and the most active BR. Immunoblotting analysis indicated that without the treatment of BL, Col-0, *TPD1 bri1-116*, and *EMS1 bri1-116* showed very low levels of non-phosphorylated BES1 (Fig. 7a, b). Interestingly, without exogenous treatment of BL, *TPD1 EMS1-GFP bri1-116*, and *TPD1 EMS1-GFP* plants showed significant accumulation of non-phosphorylated BES1, reminiscent to that of BL-treated Col-0 (Fig. 7a, b). Moreover, we compared the BES1-YFP and BZR1-YFP nuclear localization in Col-0 with those in *ems1-c1* (a knockout mutant of *EMS1* generated by using CRISPR-Cas9) (Supplementary Fig. 7), we found that the YFP signals are mainly localized in the tapetum cell nuclei in Col-0. But a significant amount of YFP signals can be detected in the cytoplasm of the somatic layer cells close to microsporocytes in *ems1-c1* background, suggesting a functional EMS1 receptor is critical to the nuclear localization of BES1 and BZR1 (Fig. 7c, d). These results demonstrate that BES1 not only can be activated by

the BRI1-BR-BAK1 signaling pathway, but also by the EMS1-TPD1-SERK1/2 pathway, regardless of the BR signaling pathway.

## Discussion

Our genetic, histological, molecular, and biochemical analyses demonstrate that BES1 family transcription factors play redundant and key roles in tapetum development. First, the *bes1-1 bzr1-1 beh1-1 beh3-1 beh4-1* quintuple mutant (*qui-1*) does not possess typical tapetum cell identity. At early stage 5 during anther development, the tapetum cell layer in *qui-1* is indistinguishable from that in Col-0. The tapetal cells in *qui-1*, however, cannot further expand at late stage 5 and are unable to stimulate the development of microsporocytes at the late anther developmental stages (Fig. 2). Tapetum marker genes and tapetum cell fate determination genes cannot be detected in these cell layers in *qui-1* (Fig. 3). As a result, *qui-1* microsporocytes fail to develop into functional pollen grains, causing male sterility (Fig. 1 and Supplementary Fig. 3). Although loss-of-function mutants of *TPD1*, *EMS1*, *SERK1/2*, *BES1* family members, and *DYT1* show similar genetic consequence regarding microsporocyte development, their tapetum developmental defects are different. *tpd1*, *ems1* and *serk1/2* lack tapetum cell layer, whereas *qui-1* and *dyt1* have a tapetum-like cell layer but they do not possess tapetum cell identity (Fig. 5). Why does *qui-1* still contain a tapetum-like cell layer? One possibility is that there are still one fully functional *BEH2* gene and partially functional *BES1* and *BEH4* in *qui-1*. Alternatively, there could exist additional branching pathways besides the BES1-mediated pathway downstream of the EMS1-TPD1-SERK1/2 signaling to regulate the development of tapetum cell layer. To clarify these issues, we generated a number of null mutants for *BES1*, *BEH2*, and *BEH4* by using a CRISPR-Cas9 approach and created the second set of quintuple mutant (*qui-2*), and the sextuple mutant with all the BES1 family members being knocked out (Supplementary Figs. 7, 8). A great portion of *qui-2* lobes show only three somatic cell layers, identical to those from *ems1*, *tpd1*, and *serk1/2* (Fig. 5d and Supplementary Fig. 8h). The remaining portion of lobes exhibit *spl*-like phenotypes (Supplementary Fig. 8l). In sextuple mutant, the majority lobes show a *spl*-like phenotype (Supplementary Fig. 8i, m). These genetic results strongly suggest that the BES1 family members are in the same signaling pathway as EMS1-TPD1-SERK1/2. Second, our quantitative RT-PCR results indicated that all the tested downstream response genes for EMS1, are also the downstream response genes for BES1 family members (Fig. 5e). Third, *bes1-D* and *bzr1-1D* can significantly suppress the male sterility of *tpd1*, *ems1*, and *serk1/2* (Fig. 6 and Supplementary Fig. 9), but cannot restore that of *dyt1* (Supplementary Fig. 10e, j, o, p). Finally, the phosphorylation and nuclear localization of BES1 are regulated by EMS1-mediated pathway (Fig. 7). In summary, our data clearly demonstrate that BES1 family members are key downstream components of the EMS1-TPD1-SERK1/2 signaling cascade in controlling the differentiation and functions of tapetum (Fig. 8).

Given the fact that *EMS1*, *TPD1*, and *SERK1/2* are all ubiquitously expressed[35–39]. If EMS1-TPD1-SERK1/2 can activate BR downstream transcription factor, BES1, why do BR signaling and biosynthetic mutants show an extremely severe defective phenotype? The activated BES1 from the EMS1-TPD1-SERK1/2 pathway should more or less compensate for the loss of the BR signaling pathway. One possible explanation is that in vegetative tissues, the EMS1-TPD1-SERK1/2 signaling pathway cannot be fully turned on due to under threshold concentrations of EMS1 and/or TPD1. Previous studies indicated that at early anther development stages (stages 2–5), the expression levels of *EMS1* and *TPD1* reach their peaks in anthers[35–37]. Possibly, only when EMS1 and TPD1 reach significant levels can the

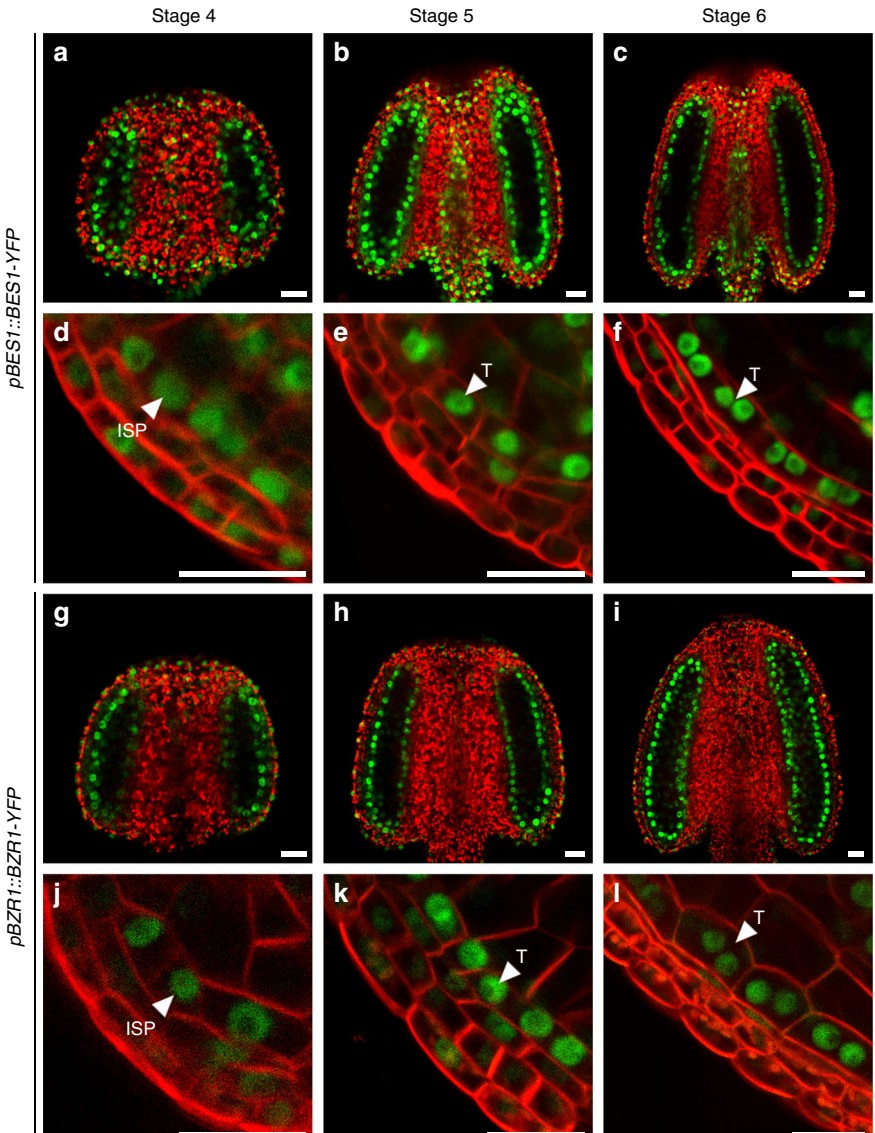

**Fig. 4** BES1-YFP and BZR1-YFP can be detected in the developing anthers. Seven-week-old *pBES1::BES1-YFP* (**a–f**) and *pBZR1::BZR1-YFP* (**g–l**) transgenic plants were used for confocal microscopic analyses. Stage 4–6 intact anthers were selected for analyzing the YFP signals. Green colors represent YFP signals and red colors are the auto-florescence of chlorophylls (**a–c**, **g–i**). To visualize detailed tissue-level localization of the YFP signal, somatic cell membranes were stained with FM4-64 (**d–f**, **j–l**). Inner secondary parietal cell layer (ISP) and tapetum (T) are indicated by arrow heads. Scale bars represent 20 μm

EMS1-TPD1-SERK1/2 signaling be initiated and the downstream signaling cascade be triggered, including the accumulation of non-phosphorylated BES1. Our quantitative RT-PCR results revealed that *BRI1* is much less expressed than *EMS1* in Arabidopsis inflorescences and siliques (Supplementary Fig. 13). In rosette leaves and cauline leaves, however, *BRI1* is more expressed than *EMS1* (Supplementary Fig. 13). Therefore, even though EMS1-TPD1-SERK1/2 and BRI1-BR-BAK1 share BES1 and its paralogs as their downstream signaling components, they may mainly function in separate territories, EMS1 functions predominantly in reproductive tissues and BRI1 plays a more significant role in the vegetative tissues. This notion is consistent with our biochemical results. For example, in Col-0 or *bri1-116*, where EMS1-TPD1-SERK1/2 is present, non-phosphorylated BES1 was undetectable. But when *EMS1* and *TPD1* were both ectopically expressed in these plants by using a relatively stronger promoter, the *BRI1* promoter, significant amount of non-phosphorylated BES1 can now be easily detected (Fig. 7).

Interestingly, some of the *35S::EMS1* transgenic lines show significantly accumulated non-phosphorylated BES1 (Supplementary Fig. 14). While none of the *35S::TPD1* transgenic lines show significant accumulation of non-phosphorylated BES1 (Supplementary Fig. 14). In *bri1* null mutants, plants are tiny, but anthers can still produce fairly normal pollen grains (Supplementary Fig. 6). In an *ems1* null mutant, vegetative tissues look indistinguishable from those of wild-type Col-0, but tapetum development is greatly impaired. As a consequence, no pollen grains can be generated.

Our genetic data suggest BES1 family members serve as downstream components of the EMS1-TPD1-SERK1/2 signaling complex and as upstream components of DYT1. For example, the expression levels of *DYT1* in *qui-1* is greatly reduced, and *bes1-D* or *bzr1-1D* cannot rescue the defective phenotypes of *dyt1* (Fig. 3 and Supplementary Fig. 10). To test whether BES1 and BZR1 can directly regulate the expression of *DYT1*, we carried out a chromatin immunoprecipitation (ChIP) analysis to investigate

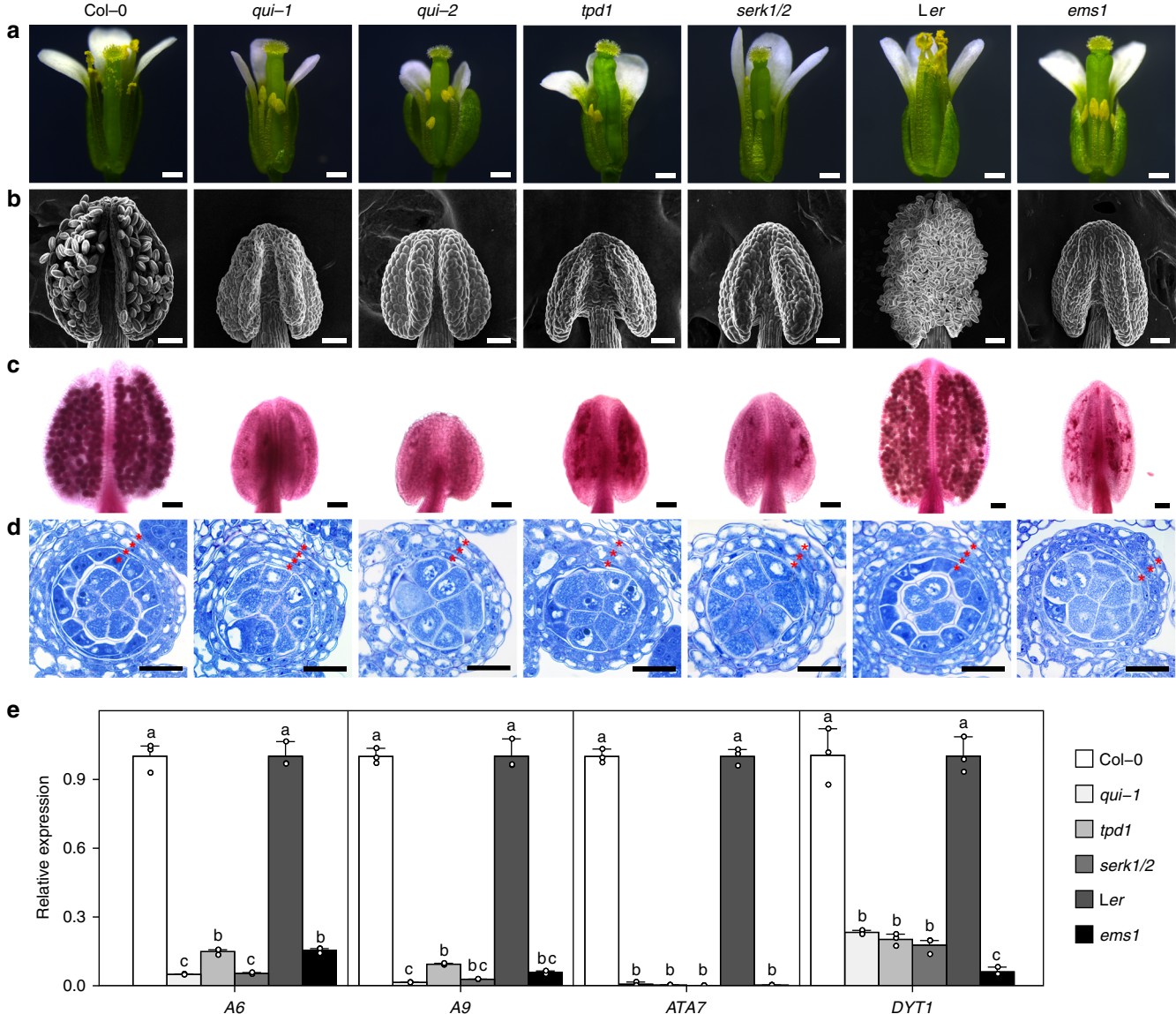

**Fig. 5** *bes1-1 bzr1-1 beh1-1 beh3-1 beh4-1 (qui-1)*, *bes1-c1 bzr1-1 beh1-1 beh3-1 beh4-c1 (qui-2)*, *tpd1*, *serk1 serk2*, or *ems1* show similar anther developmental defects. **a** Floral phenotypes of Col-0, *qui-1*, *qui-2*, *tpd1*, *serk1/2*, Ler and *ems1*. The *qui-1* and *qui-2* display shortened filaments similar to *tpd1*, *serk1 serk2*, and *ems1*. **b**, **c** SEM microscopic (**b**) and Alexander staining (**c**) images of the anthers from the plants corresponding to (**a**). *qui-1* and *qui-2* anthers cannot produce pollen grains, reminiscent to those of *tpd1*, *serk1/2*, and *ems1*. **d** Toluidine blue stained semithin sections of stage 6 anthers from plants corresponding to (**a**). All anthers of *qui-1*, *qui-2*, *tpd1*, *serk1/2*, and *ems1* show excessive microsporocytes. *qui-1* contains an impaired tapetal cell layer, whereas *qui-2* only contains three somatic cell layers, lacking the tapetal cell layer, which is identical to those of *tpd1*, *serk1/2*, and *ems1*. Number of somatic cell layers are indicated by red asterisks. **e** Quantitative RT-PCR results showing relative expression levels of tapetum marker genes in the inflorescences of various mutants and their corresponding wild-type backgrounds. Data are presented as mean and s.d. ($n = 3$). Statistically significant differences between groups were tested using One-way ANOVA followed by LSD (least significant difference) post hoc test. Different letters indicate significant difference at $P < 0.05$. Scale bars represent 0.5 mm in (**a**), 50 μm in (**b**), (**c**), and 20 μm in (**d**).

whether BES1 and BZR1 can be enriched in two E-box motifs within the *DYT1* promoter region (Supplementary Fig. 15). As a matter of fact, BZR1 showed strong enrichment in both motifs. These data suggest BES1 family members may directly regulate the expression of *DYT1*. Further detailed analyses are required to confirm such a conclusion. In addition, because a majority of anther lobes from sextuple mutant exhibit *spl*-like anther phenotypes, it is possible that BES1 family transcription factors may also regulate the expression of *SPL* in earlier anther development. As a matter of fact, a previous report indicated that BES1 can be enriched in the promoter region of *SPL*[55]. Our experiment also confirmed this result (Supplementary Fig. 15c).

Detailed analyses are also required to confirm such a scenario in the future.

During our genetic analyses, we noticed that *qui-2* and the sextuple mutant showed weak and strong *bri1* phenotypes, respectively, confirming the significance of BES1 family members in regulating the BR signaling pathway. This report, however, focuses on the regulation of BES1 family members in tapetum development. Analyses of the genetic importance of BES1 family members in regulating the BR signaling pathway was discussed in a separate report[60]. Our data presented in this report contribute to a better understanding of the EMS1-TPD1-SERK1/2-mediated tapetum developmental signaling pathway, of which the

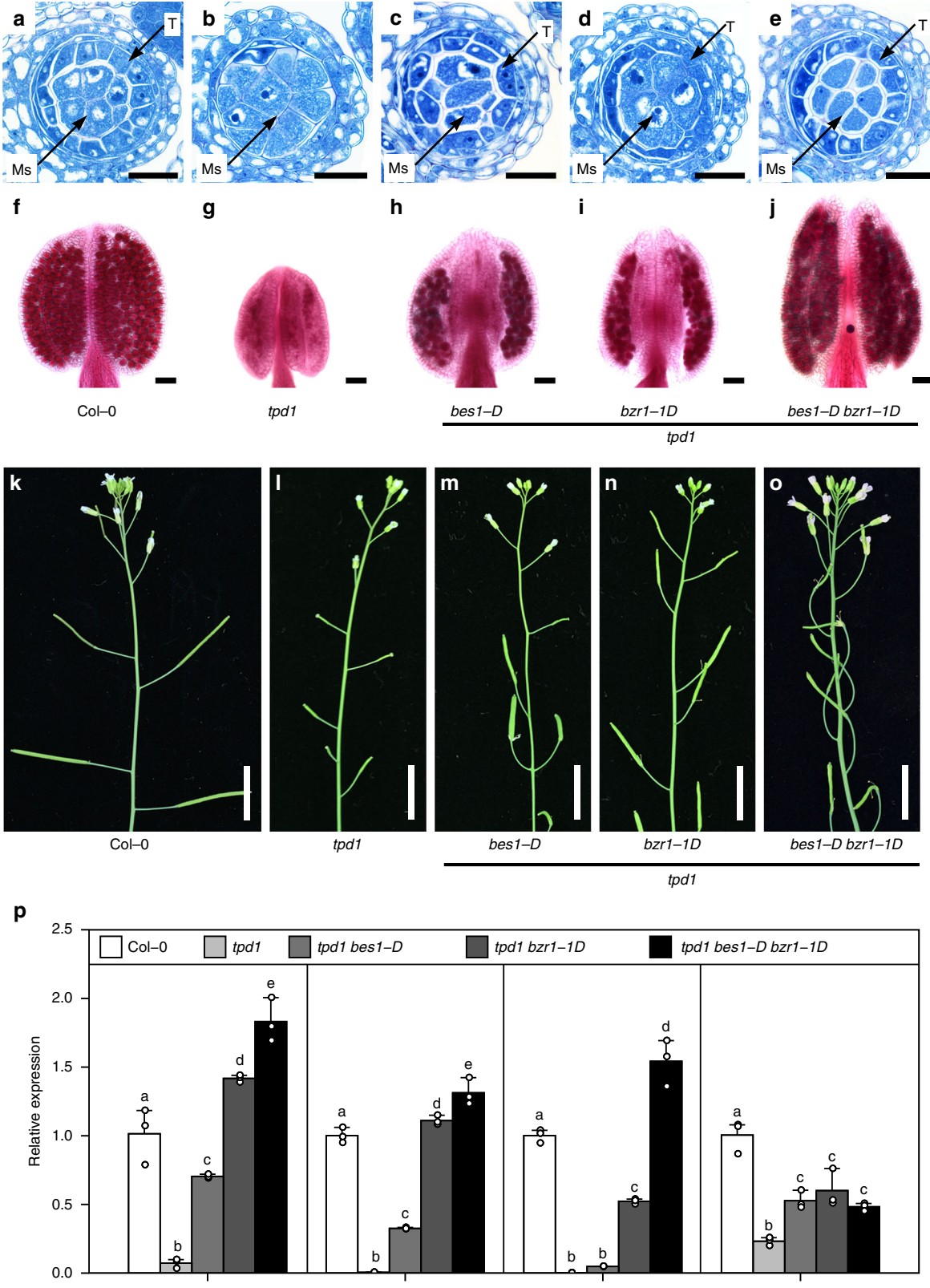

**Fig. 6** *bes1-D* and *bzr1-1D* can partially suppress anther developmental defects and male sterility of *tpd1*. **a–e** Toluidine blue stained anther semithin sections of Col-0 (**a**), *tpd1* (**b**), *tpd1 bes1-D* (**c**), *tpd1 bzr1-1D* (**d**), and *tpd1 bes1-D bzr1-1D* (**e**). **f–j** Alexander stained anthers of Col-0 (**f**), *tpd1* (**g**), *tpd1 bes1-D* (**h**), *tpd1 bzr1-1D* (**i**), and *tpd1 bes1-D bzr1-1D* (**j**). **k–o** Inflorescences of Col-0 (**k**), *tpd1* (**l**), *tpd1 bes1-D* (**m**), *tpd1 bzr1-1D* (**n**), and *tpd1 bes1-D bzr1-1D* (**o**). Tapetum layer (T), and microsporocytes (Ms) are indicated by arrows. **p** Quantitative RT-PCR analyses to show relative gene expression levels of tapetum marker genes, *A6*, *A9*, *ATA7*, and *DYT1* in the inflorescences of Col-0, *tpd1*, *tpd1 bes1-D*, *tpd1 bzr1-1D*, and *tpd1 bes1-D bzr1-1D*. Data are presented as mean and s.d. ($n = 3$). Statistically significant differences between groups were tested using one-way ANOVA followed by LSD (least significant difference) post hoc test. Different letters indicate significant difference at $P < 0.05$. Scale bars represent 20 μm in (**a–e**), 50 μm in (**f–j**), and 1 cm in (**k–o**)

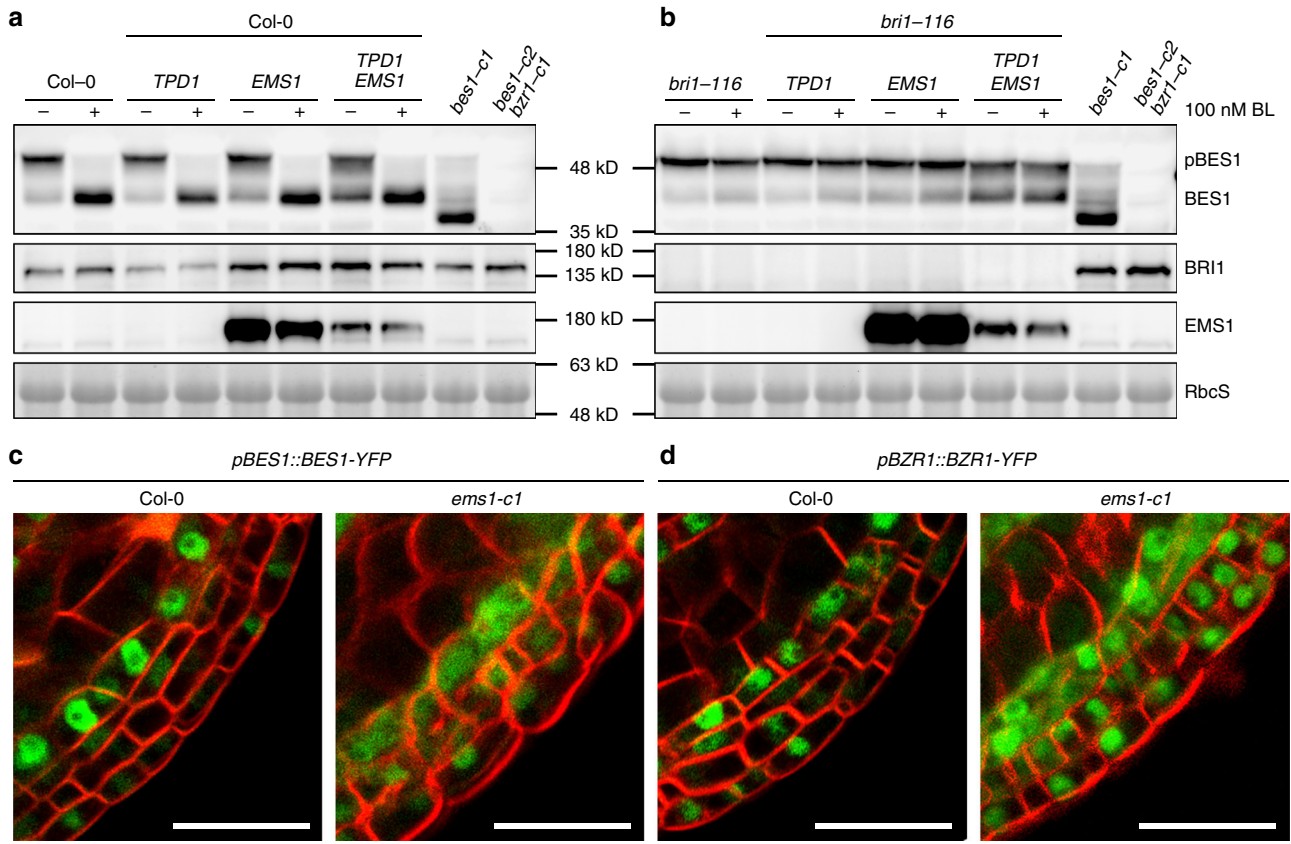

**Fig. 7** BES1 can be activated by the EMS1-TPD1-SERK1/2 signaling pathway. **a** Immunoblotting analyses using transgenic plants harboring *pBRI1::TPD1* (*TPD1*), *pBRI1::EMS1-GFP*(*EMS1*), or both in Col-0 indicated that the expression of *TPD1* and *EMS1* can lead to the accumulation of non-phosphorylated BES1. **b** The TPD1- and EMS1-induced accumulation of non-phosphorylated BES1 is independent of the BR signaling. Homemade anti-BES1 antibody was used to detect phosphorylated (p-BES1) or non-phosphorylated BES1 (BES1). *bes1-c1*, *bes1-c2 bzr1-c1* were used as negative controls for the anti-BES1 antibody. Coomassie Brilliant Blue stained rubisco protein was used as a loading control. EMS1-GFP and BRI1 immunoblotting analyses were carried out by using anti-GFP or anti-BRI1 antibody to confirm the accumulation of EMS1-GFP and the *bri1-116* background, respectively. **c, d** Nuclear localization of BES1 or BZR1 in the somatic cell layer closest to the microsporocytes is partially dependent on the presence of EMS1. Scale bars represent 20 μm

downstream signaling components are largely uncharacterized. To our knowledge, this is also the first piece of evidence to show that BES1 family members can be shared by two distinctive RLK-mediated signaling pathways. It is likely that additional signaling pathways may also use BES1 family members as their downstream signaling components.

## Methods

**Phylogenetic analysis**. Full-length amino acid sequences of BES1 family members were obtained from The Arabidopsis Information Resource (TAIR, https://www.arabidopsis.org/). Multiple sequence alignment was performed on MEGA 7.0.14 program (https://www.megasoftware.net/home) using MUSCLE method. The unrooted neighbor-joining phylogenetic tree was generated on the same program by using the Jones–Taylor–Thornton (JTT) model, and was gamma-distributed (gamma = 0.97) with 1000 bootstrap replicates.

**Plant materials and growth conditions**. The *Arabidopsis thaliana* Columbia (Col-0) and Landsberg *erecta* (L*er*) ecotypes were used in this study. Mutants of *BES1* and its paralogs were obtained from ABRC with accession numbers: *bes1-1* (SALK_098634), *bzr1-1* (GK_857E04), *beh1-1* (GK_432C09), *beh3-1* (SALK_017577), *beh4-1* (SALK_055421). Primers for genotyping are listed in Supplementary Table 1. The other Arabidopsis lines are described previously: *bri1-701*[53], *bri1-116*[53], *serk1/2*[53], *cpd*[54], *bin2-1*[16], *bes1-D*[21], *bzr1-1D*[20], *tpd1*[37], *ems1*[36], *bam1/2*[58], *spl*[57], *rpk2*[59], and *dyt1*[43]. After vernalizing at 4 °C for 2 days, all the described wild-type and mutant seeds were planted and grown in soil at 22 °C under 16 h-light/8 h-dark photoperiod condition. For western blot analysis, seeds were grown on half-strength Murashige and Skoog (1/2 MS) agar plates, supplemented with 1% (w/v) sucrose under the same condition described above.

**Generation of constructs and transgenic plants**. For complementation, promoter (~1.5 kb) plus genomic sequences of *BES1* and its paralogs (except *BEH3*) were cloned into a Gateway™ Entry vector, then subcloned into a binary destination vector pBIB-BASTA-GWR. Coding sequence of *BEH3* was cloned into the Gateway entry vector, then subcloned into a pBIB-BASTA-pUBQ10-GWR vector, derived from a pBIB-BASTA-35S-GWR binary vector. For expression analysis, the entry vectors of *BES1* and its paralogs were subcloned into a pBIB-BASTA-GWR-YFP destination vector. For ectopic expression, coding sequences of *EMS1* and *TPD1* were cloned into entry vector and then subcloned into pBIB-BASTA-35S-GWR, pBIB-KAN-pBRI1-GWR-GFP, and pBIB-HYG-pBRI1-GWR vectors, respectively. The constructs were then transformed into Col-0, *bri1-116*[+/−], and quintuple mutants through the floral dip method[61].

**Photographing and microscopy**. For phenotypic observation, photos of 7-week inflorescences of all genotypes were taken with a Canon EOS 70D Camera. Pictures of flowers from each genotype were taken with a Leica M165C Stereomicroscope. For scanning microscopic analyses, anthers were removed from fully opened flowers and carefully set on the sample preparation platform and quickly frozen in liquid nitrogen, the samples were then immediately transferred into the chamber of a Hitachi S-3400N scanning electron microscope before the images were taken. For pollen viability analysis, flower buds at development stage 12 were fixed with Carnoy's fixative containing 60% ethanol, 30% chloroform, and 10% acetic acid for at least 2 h. Individual anthers were obtained and stained with Alexander's solution for 30 min at 25 °C, the anthers were visualized and photographed with a Leica DM6000 B microscope. For anatomic studies of anthers, entire inflorescences were fixed in FAA fixative and dehydrated with a series of graded ethanol. After dehydration, the inflorescences were embedded in Technovit 7100 resin. Cross sections were made by a rotary microtome with thickness of 2 μm. The sections were then stained with 0.5% toluidine blue and the images were taken using a Leica DM6000 B microscope. For expression pattern analyses of YFP inframe-fused BES1 and its paralogs, anthers from stages 4–6 were mounted in water and were

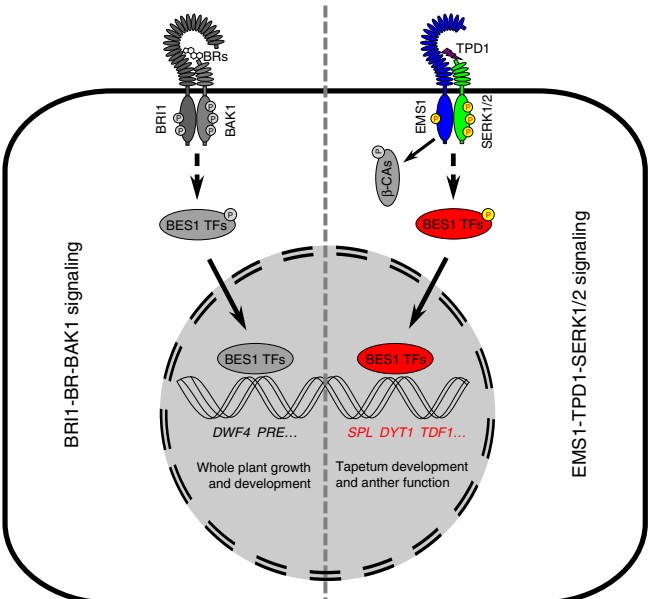

**Fig. 8** A current model showing that BES1 family members (or BES1 transcription factors, BES1 TFs) are essential downstream regulatory components in the TPD1-EMS1/SERKs pathway. In previous studies it was thought that BES1 family members play fundamental roles in regulating BR signaling pathway. A previous report proposed that EMS1 regulates tapetum development through interaction and activation of a group of beta carbonic anhydrases (β-CAs). In this study we suggest that BES1 family members can be activated by EMS1-mediated signaling to regulate tapetum formation. Our ChIP analyses suggest that BES1 and BZR1 can directly bind to the promoter regions of a number of tapetum developmental genes, such as *SPL*, *DYT1*, and *TDF1*. In addition, phenotypic similarity between the sextuple mutant and *spl* implies that BES1 family members may control *SPL* expression in earlier anther development. The detailed mechanism remains to be elucidated

observed under a Leica TCS SP8 laser scanning confocal microscope. YFP and chlorophyll signals were excited at a 514-nm laser wave length. The emission was captured using PMTs set at 517–543 and 620–690 nm, respectively. In addition, for tapetum localization of BES1, anthers were mounted and stained on slides with FM4-64 solution for at least 30 min before visualizing and photographing. YFP signal was excited and captured as described. The FM4-64 signal was excited at a 514-nm laser wave length and captured with PTM set at 620–680 nm.

**RT-PCR and qRT-PCR**. RT-PCR was used to determine the target gene expression in mutants and transgenic plants. qRT-PCR was used to evaluate the expression levels of genes that either function in or used as markers during anther development. Total RNAs were extracted from rosette leaves (RT-PCR) and inflorescences before stage 12 (qRT-PCR) using an RNAsimple Total RNA Kit (TIANGEN). cDNA was generated with the PrimeScript™ 1st Strand cDNA Synthesis Kit (Takara). For RT-PCR, genes were amplified from 100 ng total RNA reverse transcripts. For qRT-PCR, genes were amplified from 40 ng total RNA reverse transcripts. All primers used are listed in Supplementary Table 1. SYBR Premix Ex Taq II (Takara) was used in PCR reaction on a StepOnePlus Real-Time PCR System (Applied Biosystems™). All experiments were performed in triplicate.

**RNA in situ hybridization**. Inflorescences were fixed with ice-cold 4% (w/v) paraformaldehyde in glass vials on ice. After fixation in vacuum ambiance, samples were dehydrated in a series of graded ethanol. Fully dehydrated samples were stained with Eosin Y in absolute ethanol and then were cleared with graded Histo-clear/ethanol solutions. When the ethanol was replaced by pure Histo-clear, the samples were transferred into a 60 °C oven and the Histo-clear was gradually infiltrated with Paraplast (Leica). Then the fully infiltrated samples were embedded into Paraplast for sectioning. Eight-micrometer-thick cross sections on slides were dewaxed and treated with proteinase K. In vitro RNA probe synthesis, hybridization and signal detection were performed according to the manual of the DIG RNA labeling kit (Roche; 11175025910). Primers used were listed in Supplementary Table 1. Images were taken by using a Leica DM6000 B microscope.

**Western blot analysis**. After genotyping, 2-week-old double transgenic plants in *bri1-116* background were treated with or without 100 nM BL (Sigma) for 90 min. Total proteins were extracted with 2 × SDS buffer containing 125 mM Tris (pH 6.8), 4% (w/v) SDS, 20% (v/v) glycerol, 20 mM DTT, 0.02% (w/v) bromophenol blue. Protein extracts equivalent to 7.5 mg seedlings were resolved on 7% (for GFP and BRI1) and 11% (for BES1) Bis-Tris SDS-PAGE gel and transferred onto a nitro-cellulose membrane. After blocking with 5% milk solution, the membranes were incubated with primary antibodies against GFP (1:3000, Roche, 11814460001), BRI1 (1:2000, Agrisera, AS121859), and BES1 (1:5000, homemade), respectively, and then the corresponding HRP-conjugated secondary antibodies (1:5000, Abmart, anti-mouse (M21001) for GFP, and anti-rabbit (M21002) for BES1 and BRI1). The signals were revealed by a JustGene ECL Plus (CLINX) mixture and were detected by a Tanon 5200 chemiluminescence image analysis system.

**CRISPR/Cas9**. All CRISPR/Cas9 mutants mentioned were generated by using the egg cell-specific promoter-controlled (EPC) CRISPR/Cas9 system as previously reported[62,63]. For these genes, Target 1-(gRNA-Sc)-(U6-26t)-(U6-29p)-Target 2 were amplified from a pCBC-DT1T2 vector template by using the primers including the sgRNA sequence. The PCR fragments were transformed into a pHEE401E binary vector through a restriction–ligation method. The constructs were then transformed into corresponding Arabidopsis genotypes using *Agrobacterium*-mediated floral dip method[61]. The transgenic lines were confirmed by PCR and sequencing, and were crossed with their backgrounds for separation of the vector. The sequence information of targets and primers is listed in Supplementary Table 2.

**ChIP assay**. Chromatin immunoprecipitation was performed according to a previous report with minor modifications[64]. Briefly, 2 g inflorescence materials from 6-week-old Col-0, *pBES1::BES1-YFP*, and *pBZR1::BZR1-YFP* plants were cross-linked in sucrose-formaldehyde buffer (0.4 M sucrose, 10 mM Tris-HCl (pH 8), 1 mM PMSF, 1 mM EDTA, 1% formaldehyde). Nuclei were isolated with nuclei isolation buffer (0.25 M sucrose, 15 mM PIPES (pH 6.8), 5 mM MgCl₂, 60 mM KCl, 15 mM NaCl, 1 mM CaCl₂, 0.9% Triton X-100, 1 ×proteinase inhibitor) and then were lysed with lysis buffer (50 mM HEPES (pH 7.5), 150 mM NaCl, 1 mM EDTA, 1 mM PMSF, 1% SDS, 0.1% Na deoxycholate, 1% Triton X-100, 1 × proteinase inhibitor). The lysates were sonicated to generate DNA fragments with an average size of 500 bp. After 10-times dilution with dilution buffer (50 mM HEPES (pH 7.5), 150 mM NaCl, 1 mM EDTA, 1 mM PMSF, 0.1% Na deoxycholate, 0.4% Triton X-100, 1 × proteinase inhibitor), the solubilized chromatin was immuno-precipitated by an agarose-conjugated anti-GFP antibody. The coimmunoprecipitated DNA was analyzed by quantitative PCR as described above. The primers are listed in Supplementary Table 1.

**Statistical analysis**. All values are presented as mean and standard deviation (s.d.). The significance of differences for pairwise comparisons was estimated by two-tailed student's *t*-tests. The statistical significance of differences between groups were compared on SPSS statistics program using one-way ANOVA followed by least significant difference post hoc test at $P < 0.05$.

**Reporting summary**. Further information on research design is available in the Nature Research Reporting Summary linked to this article.

## Data availability

All additional data needed to evaluate the conclusions in the paper are present in the supplementary materials, including Supplementary Figs. 1 to 15 and Supplementary Tables 1 and 2. The source data underlying Figs. 3g, 5e, 6p, 7a and b, and Supplementary Figs. 3n, 8g–i, k–m, 11–14, and 15b, c are provided as a Source Data file. Other relevant data are available from the corresponding author upon a reasonable request.

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

## Acknowledgements

We thank Hong Ma from Penn State University for the gift of *ems1* seeds and Guang Wu from Shanxi Normal University for *EMS1* and *TPD1* transgenic plants; JL lab member Yao Xiao for preparation of the primary antibody against BES1; Liang Peng, Liping Guan, Yang Zhao, Yuhong Niu, Haiyan Li, and Yahu Gao (Core Facility for Life Science Research, Lanzhou University) for technical assistance. This Work was supported by the National Natural Science Foundation of China (31530005, 31720103902, 31470380), the Ministry of Education (113058A and NCET-12-0249), the 111 Project of the State Administration of Foreign Experts Affairs (B16022), and the Gansu Provincial Science and Technology Department (17ZD2NA015-06 and 17ZD2NA016-5).

## Author contributions

J.L. supervised the entire project. J.L., W.Y.C., and M.H.L. designed the experiments and analyzed the data. W.Y.C. and M.H.L. carried out most of the experiments. Y.Z.W., P.A.W., Y.W.C., M.Z.L., and R.S.W. helped for some of the experiments; X.P.G. provided suggestions for the paper. J.L. and M.H.L. wrote the paper.

## Additional information

**Competing interests:** The authors declare no competing interests.

