## [Peer Review File · Nature Communications]

RESPONSE TO THE COMMENTS FROM THE REVIEWERS:

Reviewer 1

Comments:

The revised manuscript from Chen and Lv et al. has addressed the comments on the previous version and is greatly improved.

The following minor comments should be addressed.

1. In line 2 of the abstract, did the authors mean “(BR) response gene expression”, or perhaps “(BR)-mediated response and gene expression”? The words “mediated response gene expression” seem awkward.

Our response: We have modified the sentence.

2. In line 2 from the bottom of the abstract, please change “BES1 family” to “the BES1 family”

Our response: We have added “the” before “BES1 family”.

3. Lines 93, 193, 355, maybe other places, please replace “pollens” by “pollen grains” because “pollen” is usually used as the singular form to refer to the male gametophyte as a type of structure, but not as individual grains.

Our response: We have changed “pollens” to “pollen grains” in all the places throughout the text.

4. Line 272, it might be better to change “their family” to “other family”.

Our response: We have changed “their family” to “other family”.

5. Line 345, it would be better to changes the words to “two additional insertions of 8-bp and 24-bp”

Our response: We have changed as the reviewer suggested.

6. Line 411, please change “obviously” to “obvious”.

Our response: Changed.

7. Line 415, please remove “double”, because rpk1 was a single mutant.

Our response: Removed.

8. Please check other parts to see whether there might be other minor problems.

Our response: We have checked the entire text and corrected some minor problems.

9. In the refs 3, 32, 33, 42, 53, 54, it is not clear whether the authors intended to have the first letter of most words of the title capitalized.

Our response: We have fixed the problems the reviewer identified and also additional problems in the references.

10. The authors indicated in their response to the second reviewer’s comments that they performed in situ RNA hybridization experiments for BEH2, but did not report the results because the corresponding sense probe showed strong signals.

Although it is common to use the sense probe as a negative control in in situ experiment, this is not the only possible negative control. The sense probe is used because it can be easily generated from the same cloned fragment of the gene being tested. However, sometimes the sense probe happens to hybridize to something by chance. In this case, another sense probe of a different gene can be used as the negative control, because there is no strong reason that the sense probe of the same gene is better than other sense probes.

Because the expression information of BEH2 is helpful, it is suggested here that this results be included, with a different sense probe.

Our response: This is a good suggestion. We will use such technique in some other occasions. But in this manuscript, we have tried different anti-sense probes and sense probes for *BEH2* and we found all the anti-sense probes we tried did not produce specific signals (the signals were every where). Therefore, different sense probes will not help us to believe that the signals seen from in situ hybridization represent the true expression patterns of *BEH2*. In addition, in our quintuple mutants, *BEH2* gene is not knocked out, suggesting the minor roles in regulating tapetum development. Alternatively, we provided the expression patterns of *BEH2* using *pBEH2::GUS* transgenic plants and confirmed the low expression of *BEH2*. For the aforementioned reasons, we decided not to include the *in-situ* hybridization results of *BEH2* in the revised manuscript.

Reviewer 2

Comments: The authors have addressed all of my concerns. This work is of a high interest and it should be published in Nature Communications.

In the revised manuscript Chen et al. present new data which resolve most of the issues raised by reviewers. Their findings about defects of tapetum development in quintuple BES1 family mutant *qui-1* are further corroborated by isolation of one additional quintuple mutant - *qui-2*, and even sextuple CRISPR/Cas9 generated mutant. Their findings clearly demonstrate involvement of BES1 family of TFs in anther development and more specifically their role in cell fate specification of the tapetum layer, as downstream components of EMS1-TPD1-SERK1/2-mediated signaling pathway.

The question of BEH2 gene expression pattern has been addressed, maybe not in sufficient detail, but pBEH2:GUS lines show low level of expression in the inflorescence as expected, which explains the sterility of the *qui-1* mutant. Additionally, analysis of the newly isolated sextuple mutant clearly shows importance of all BES1 family members for anther development.

In my view, improved complementation of *tpd1* mutant phenotype by simultaneous expression of *bes1-D* and *bzr1-D* is a strong indicator of involvement of BES1 family TFs in EMS1-TPD1 signaling pathway.

There are no additional questions from reviewer 2.